# Fucoidans and Bowel Health

**DOI:** 10.3390/md19080436

**Published:** 2021-07-30

**Authors:** Jin-Young Yang, Sun Young Lim

**Affiliations:** 1Department of Biological Sciences, Pusan National University, Busan 46241, Korea; jyyang99@pusan.ac.kr; 2Division of Convergence on Marine Science, Korea Maritime & Ocean University, Busan 49112, Korea

**Keywords:** fucoidan, intestinal function, gut microbiota, inflammation, immunity

## Abstract

Fucoidans are cell wall polysaccharides found in various species of brown seaweeds. They are fucose-containing sulfated polysaccharides (FCSPs) and comprise 5–20% of the algal dry weight. Fucoidans possess multiple bioactivities, including antioxidant, anticoagulant, antithrombotic, anti-inflammatory, antiviral, anti-lipidemic, anti-metastatic, anti-diabetic and anti-cancer effects. Dietary fucoidans provide small but constant amounts of FCSPs to the intestinal tract, which can reorganize the composition of commensal microbiota altered by FCSPs, and consequently control inflammation symptoms in the intestine. Although the bioactivities of fucoidans have been well described, there is limited evidence to implicate their effect on gut microbiota and bowel health. In this review, we summarize the recent studies that introduce the fundamental characteristics of various kinds of fucoidans and discuss their potential in altering commensal microorganisms and influencing intestinal diseases.

## 1. Introduction

Living animals, including humans, ingest food and digest it to obtain essential nutrients for maintaining physical strength. Nutrient absorption produces energy that can be used in different metabolic systems, mainly growth [1], development [2], and reproduction [3]. Thus, a well-balanced diet is very crucial to sustain not only host health but also host immunity against invading pathogens. In this context, food additives that boost the immune system can provide additional benefits to the host. Recently, with the growing concern towards healthcare, the use of artificial chemical compounds, including antibiotics is increasingly being avoided. Thus, researchers have been looking for naturally derived substances that can be beneficial for human health without any side effects. Of these, bioactive compounds from seaweed sources, including micro- and macro-algae, are promising candidates with multiple beneficial functions, including antioxidant [4], anti-inflammation [5] and anti-cancer activity [6].

“Fucoidans” refer to fucose-containing polysaccharides with sulfate groups, extracted from brown seaweed species such as *Fucus vesiculosus*, *Cladosiphon okamuranus*, *Laminaria japonica* and *Undaria pinnatifida* [7]. It is well-known that fucoidans are present in the seaweed cell wall to sustain cell membrane stability and protect their structure against dehydration [8]. The structure of fucoidans has been correlated with their biological and immunological activities, such as antioxidant, anti-coagulant, anti-thrombotic, anti-inflammatory, anti-viral, anti-lipidemic, anti-metastatic, anti-diabetic and anti-cancer effects [9]. Although fucoidan extracts have been widely used in various health care products including food supplements [10] and cosmetics [11], their bioactivities were reported to be sensitive to alterations in their structural composition. Moreover, understanding their functional properties are complicated since it is dependent on fucoidan extraction methods and the different seaweed species [12]. Some studies have demonstrated that functional characteristics of fucoidan extracts are closely linked to their structural formula, compositions of sugars, sulfate content and sulfate group positioning [9,13]. Therefore, more knowledge about structural and chemical characteristics of fucoidan is essential to better understand functional properties of the fucoidan.

The intestine is the most dynamic and the largest compartment of the immune system where nutrient digestion and absorption take place. Moreover, it plays a critical role in host immunity to maintain intestinal homeostasis through several interactions between immune cells and microbes, and the immune cells actively monitor, recognize, and differentiate food antigens from external antigens including pathogens [14]. Thus, many studies have attempted to modulate the intestinal immune system by employing immunostimulants (or immune modulators). It is well-understood that the stimulants could induce intestinal immunity by stimulating pattern recognition receptor signaling pathways in antigen-presenting cells and epithelial cells, resulting in the production of certain cytokines or chemokines [15,16]. Fucoidan, a candidate immunostimulant, may improve host intestinal immunity, considering its immunological role as a functional supplement. Breast cancer-bearing rats fed a fucoidan supplemented diet have been reported to exhibit significantly improved intestinal barrier functions and altered diversity and composition of intestinal flora, showing that fucoidan can be a potential protective agent against breast cancer [17]. A recent study on zebrafish [18] demonstrated that fucoidan extracted from Okinawa mozuku (*Cladosiphon okamuranus*) altered the composition of intestinal microbiota and the number of intestinal neutrophils and macrophages. In addition, fucoidan treatment with a meal significantly decreased the expression of pro-inflammatory genes (*IL1b*) in the adult zebrafish intestine, highlighting the potential immunomodulatory functions of fucoidan.

Therefore, in this review, we provide an overview of the diversity of the structural and chemical composition of fucoidans and their possible immunological effects on the intestine accompanied by gut microbiota alteration. Furthermore, we suggest the prospective application of fucoidan in modulating host intestinal diseases.

## 2. Influence of Fucoidans on Intestinal Bowel Function

Owing to its anatomical position, the gastrointestinal (GI) tract is a specialized barrier to maintain intestinal homeostasis by physically separating the intestine from the external environment [19]. In addition to nutrient uptake, the GI tract plays an important role in maintaining intestine immunity by identifying “Good” or “Bad” foreign antigens and enabling their interaction with diverse intestinal cells [20]. Notably, Xue, et al. [17], Zuo, et al. [21] suggest that dietary fucoidans can directly stimulate intestinal cells. Here, to understand physiological function of dietary fucoidans, we discuss how fucoidans can affect intestinal non-immune cells and immune cells (Table 1).

### 2.1. Effect on “Non-Immune Cells”

The intestinal epithelial barrier includes various types of cells, such as enterocytes, goblet cells, Paneth cells, and enteroendocrine cells, which are derived from multipotent intestinal stem cells (ISCs). A recent study showed that dietary components can also affect the differentiation of intestinal progenitor cells [22]. Lgr5^+^ ISCs can only recognize digested fructose and further differentiate into absorptive (i.e., enterocytes) or secretory progenitors (i.e., tuft cells, goblet cells, and Paneth cells), respectively. In addition, the intake of carbohydrates and proteins ameliorates intestinal epithelial injury by increasing membrane permeability [23]. Enterocytes line the surface of the epithelium in the small and large intestines, joined to each other by tight junctions. They are known to interact with nutrients including polysaccharides through antigen uptake and endocytosis [24]. Another study demonstrated that fucoidan extracted from *Sargassum cinereum* reduced the growth of the colon cancer cell line Caco-2, which has characteristics similar to enterocytes with a brush border epithelial layer [25]. In a study using human enterocyte-like HT-29-luc cells treated with extracts from *Undaria pinnatifid*, higher superoxide anion radical scavenging capacities and increased cell viability were shown compared to control groups [26]. Other studies on Caco-2 cell lines [27] showed that the eggshell membrane protein from chitosan/fucoidan nanoparticles reduced NO products and expression levels of TNF-α and IL-6, as well as having increased the paracellular permeability of fluorescein isothiocyanate-dextran in IECs, suggesting the importance of increasing bioavailability of fucoidan extracts to optimize their efficacy on immune response. Furthermore, a study on mice [28] found that fucoidan extracts could inhibit cryptosporidiosis by reducing the adhesion of *Cryptosporidium parvum* oocysts in the intestinal epithelial cells. These finding demonstrate the cytotoxic roles of fucoidan with potential applications in cancer prevention. In a mouse study [29], mucins secretion by mucous/goblet cells was considerably increased in the ileum and feces of tumor necrosis factor (TNF) receptor-associated factor 3-interacting protein 2 (Traf3ip2) mutant mice fed a diet containing fucoidan from *Cladosiphon Okamuranus*, suggesting its protective roles against psoriasis.

### 2.2. Effect on “Immune Cells”

Fucoidan has also been reported to modulate immune cell counts and their functions. A study employed immunohistochemistry to identify fucoidan-positive cells and their cell types in the small intestine of rats fed 2% fucoidan. The authors detected fucoidan and ED1 (macrophage marker) double-positive cells, indicating that intestinal macrophages may be the main cell type to internalize fucoidan [30]. A recent study showed that fucoidan treatment also affects the production of intracellular reactive oxygen species and recruitment of macrophages and neutrophils in LPS-induced RAW 264.7 murine macrophage cell lines and zebrafish larvae [31]. In the case of dendritic cells (DCs), significantly lower expressions of MHCII and CD86 was observed in bone marrow-derived DCs of non-obese diabetic mice fed fucoidan compared to those in the control group [32]. This suggests that fucoidan conserves DCs in an immature state, exerting immune tolerance in mice. Fucoidan also prevented the nuclear translocation of nuclear factor (NF)-κB p52 in B cells, and stimulated murine B cell proliferation for 48 h incubation after co-stimulation with interleukin (IL)-4 and anti-CD40 antibodies [33], suggesting a defensive role against immunoglobulin (Ig) E-associated diseases. Significant increase in CD8^+^ T cells and decrease in the ratio of CD4^+^/CD8^+^ T cells were observed in the spleen of mice fed with high molecular weight fucoidan compared to those in the control group, suggesting that dietary fucoidan could increase cytotoxic T cell response [34]. Another group found that oligo-fucoidan-treated mononuclear cells isolated from the peripheral blood of asthmatic patients showed larger Th1 and Treg populations and increased IL-10 production than the control group, indicating potential anti-inflammatory functions of the fucoidans [35]. A study on mice [36] found that fucoidan can increase CD40, CD80, CD86, IL-6, IL-12 and TNF-α in spleen DCs and promote T cell proliferation of CD4^+^ and CD8 T^+^ cells, indicating that fucoidan can stimulate Th1 immune responses. A recent study on human mononuclear U937 cells [37] found that fucoidan extract from *Fucus vesiculosus* L. showed higher inhibition of MAPK p38 than those of SB203580, a potent p38 MAPK inhibitor, and greater inhibition of COX-2 enzyme activity than the synthetic non-steroidal anti-inflammatory drug indomethacin with increasing the fucoidan amounts. Ruslan Medzhitov at Yale University recently demonstrated transcriptional changes in γδ T cells induced by a high-carbohydrate diet, and the regulatory functions of these cells are presumed to modulate carbohydrate transcriptional programs, including monosaccharide transporters, by controlling IL-22 production [38]. Together, these findings suggest that dietary nutrients, such as fucoidan, alter the population of gut immune cells containing macrophages, DCs, B cells, and T cells.

## 3. Fucoidan Structure

Although there is considerable variation among different algal species in terms of structural and chemical composition of fucoidans, they generally comprise the (1→3)-linked α-L-fucopyranosyl backbone structure, and occasionally both (1→3)-linked and (1→4)-linked α-L-fucopyranosyl structures [12] (Figure 1). Fucoidans, produced by the four major orders of brown algae (Chordariales, Laminariales, Ectocarpales and Fucales), appear to have distinct chemical and structural compositions but similar backbone structures, providing them with the ability to interact with various proteins, resulting in a wide range of biological activities [39]. For example, the positioning of sulfate groups differs among fucoidans from different algal species, which determines their structure and functional capabilities [40]. In fucoidans from brown seaweeds of the order Fucales, such as *Fucus serratus* L., the sulfate groups appear at the C-2 and C-4 positions, whereas in fucoidans from *Cladosiphon okamuranus* (Chordariales) the sulfate groups are at the C-2 position [41,42]. Fucoidans from *Chorda filum* (Laminariales) are sulfated at the O-2 and O-4 positions, while those from *Ascophyllum nodosum* (Fucales) are mainly sulfated at the O-2 position, and in a few cases at O-3 and O-4 positions [43,44]. Concerning their chemical composition, most fucoidans are hetero-polysaccharides that contain a fucose-containing residue, but also selectively contain different sugars such as galactose, xylose, mannose, and glucose [9], and acids such as acetate and uronic acids [45,46]. Furthermore, several studies have indicated that extraction methods and seaweed harvest time could influence fucoidan chemical composition [47]. Although the categorization of fucoidan is not simple due to the difficulties mentioned above, we summarize the previous reports on fucose-containing macroalgae based on its structure (Table 2).

## 4. Functional Effects of Fucoidans

### 4.1. As an “Energy Sources”

Fucoidan as a bioactive compound with high molecular weight and sulfated polysaccharides [64]. Fucoidan extracts contain many sugars including fucose, galactose, xylose, mannose, and glucose [9]. While fucose is generally rich in most brown seaweed species, the proportions of the saccharides vary among different species [65] (Table 2). Microvilli and plasma membranes of enterocytes play a critical role in the digestion and absorption of fucoidan [30]. There might be three possible routes of fucoidan oligosaccharides through intestinal epithelium. Many digestive enzymes including amylase, protease and lipase are attached to the plasma membrane and help in breakdown of fucoidans to the constituent monosaccharides [66,67]. These compounds may infiltrate enterocytes by active transport via Na^+^-dependent glucose transporters [68]. In addition, they could be directly recognized and transported into the epithelium by endocytosis of IECs [69]. It is also assumed that they are absorbed as short-chain fatty acids (SCFAs) by the fermentation by-products from fucoidan [70]. The transported nutrients from fucoidan as energy sources could enhance intestinal barrier integrity and maintain intestinal homeostasis by interacting with various intestinal cells (Figure 2).

### 4.2. As an “Immune Regulators”

Dietary fucoidans have been reported to have enhanced bioactivity and play pivotal roles in improving the gut health of animals, including humans [71], mice [72], livestock [73], and fish [74] (Table 3). A study on newly weaned pigs [73] indicated that fucoidan-supplemented diets significantly increased intestinal villous height and the ratio of villus height to crypt depth compared to those of the control group. This indicates that fucoidan can improve the intestinal health of animals by modulating intestinal morphology. In another study using pigs [75], diets containing laminarin and fucoidan derived from *Laminaria* spp. increased the coefficient of total tract apparent digestibility and decreased counts of *Escherichia coli* in the feces, suggesting that the increased digestive capacity from fucoidan diets maintains intestinal homeostasis. Dietary fucoidan can also enhance gut immune functions through immunomodulatory and anti-inflammatory effects. Furthermore, fucoidan can alleviate inflammation-related diseases, including inflammatory bowel diseases. A recent study demonstrated that mice receiving fucoidan during antibiotic treatment showed decreased levels of TNF-α, IL-1β, IL-6, and IL-10 in the colon tissue, indicating the potential protective role of fucoidan in colon health [76]. In a study using a dextran sulfate sodium salt (DSS) mouse model of acute colitis [72], mice fed with fucoidan extracts showed significant reduction in diarrhea and fecal blood compared to mice with untreated colitis mice. In the same study, the fucoidan extracts decreased the infiltration of inflammatory cells and the expression levels of inflammation-related cytokines, including TNF-α and IL-1β in the colonic tissue during DSS-induced inflammation. To effectively exert the nutritional and immunological effects of fucoidan, it is worthwhile studying pharmacokinetic and tissue distribution of fucoidan. Several recent studies have focused on how fucoidan is transformed after oral administration through the processes of absorption, distribution, metabolism, and excretion [77,78,79]. A study by Imbs, Zvyagintseva and Ermakova [77] revealed that fucoidan from *Cladosiphon okamuranus* is absorbed by intestinal epithelial cells and accumulated by liver macrophages, and also found in blood and urine at low levels, assuming that fucoidan transformation could be occurred with helps of gut microbiota by activating carbohydrate active enzymes. Other pharmacokinetic studies using fucoidan from *Fucus vesiculosus* [78] showed that after oral administration to rats, higher accumulations were observed in the kidneys, spleen, and liver. In the blood, a relatively longer absorption time was shown. Recently, scientists have made attempts to increase the bioavailability of fucoidan by forming nanoparticles or complexes. An in vitro study by Deepika, et al. [80] found that fucoidan-rutin, a new flavonoid-polysaccharide complex, could increase bioavailability and cancer cell apoptosis. Although it is not directly related, a study showed O-carboxymethyl chitosan/fucoidan nanoparticles enhance cellular uptake of curcumin in vitro, indicating that nanoparticle technique could have the potential to increase fucoidan absorption and its bioavailability [81]. Thus, further studies are needed to investigate whether the complexes or nanoparticles could increase bioavailability of fucoidan in in vivo models. Despite the potential application of fucoidan as a functional diet supplement, detailed knowledge about the mechanism of action of fucoidan is still lacking. The best suggested mechanism to date involves the downregulation of mitogen-activated protein kinase (MAPK) and NF-κB signaling pathways in immune and structural cells, including epithelial cells, which results in decreased pro-inflammatory and increased anti-inflammatory cytokines [82].

## 5. “Fucoidan-Microbiota-Intestine” Axis

### 5.1. Steady-State Condition

Fucoidan extracted from diverse brown seaweed species can beneficially influence host intestinal conditions by mediating the changes in the composition of commensal microbiota. Undaria pinnatifida (wakame) enriches *Bifidobacterium longum*, a well-known prebiotic that affects host metabolic disorders by enhancing glucagon-like peptide 1 (GLP-1) absorption in the intestine [83,84]. Mice fed with fucoidan from *Ascophyllum nodosum* showed a high proportion of *Lactobacillus* species, which modulate several host intestinal immunity-related processes, such as intestinal epithelial cell regeneration [85,86]. In addition, mice treated with dietary fucoidan exhibited increased levels of *Ruminococcaceae*, a key producer of short-chain fatty acids (SCFA) that maintain intestinal homeostasis by regulating Th1 and Treg cells [87,88]. Mekabu fucoidan can inhibit the efficacy of *Cryptosporidium parvum* to attach and colonize the intestinal epithelium, through direct binding to *C. parvum*-derived functional mediators in both humans and neonatal mice [28]. Other studies suggested that fucoidan extracted from *Cladosiphon okamuranus* also improves intestinal mucosal immunity by increasing the secretion of mucin and IgA into the lumen area [29]. These findings imply that dietary fucoidan may be a critical trigger for alteration of microbiota composition and consequently may affect the maintenance of intestinal homeostasis.

### 5.2. Disease Condition

Inflammatory bowel diseases (IBD), including ulcerative colitis (UC) and Crohn’s disease (CD), are the most common intestinal disorders. In particular, IBD is a disease with an increasing worldwide incidence rate, and is characterized by multiple inflammatory reactions of unknown cause in the intestine. Two different fucoidan preforms, the fucoidan-polyphenol complex and depyrogenated fucoidan (DPF), have been shown to ameliorate DSS-induced acute colitis through the downregulation of pro-inflammatory cytokines, leading to decreased pathohistological scores [72]. Fucoidan extracts from *Chnoospora minima* also improve host inflammation symptoms by inhibiting the nitrous oxide (NO)-mediated expression of prostaglandin (PG) E2 in zebrafish [89]. Another intestinal disease that seriously threatens human health is colorectal cancer, which is the third most common cancer worldwide. Emerging evidence suggests that fucoidan may be a prospective anti-cancer agent. Although the efficacy of fucoidan uptake in intestinal epithelial cells varies based on its molecular size, fucoidan extracted from *Cladosiphon okamuranus* has been shown to enhance the survival rate in mice of a colorectal tumor-bearing model [90]. Fucoidan can control cell viability and the cellular cycle by downregulating the insulin-like growth factor (IRF)-I receptor via the Ras/Raf/ERK pathway [91]. Fucoidan extracted from *Sargassum cinereum* enhances reactive oxygen species (ROS) production and, consequently, inhibits cell proliferation by increasing the permeability of the mitochondrial membrane and the efficacy of apoptosis in Caco-2 cells [25]. In addition, fucoidan has been reported to inhibit the progression of lymphocyte tissue infiltration by directly suppressing the activity of matrix metalloproteases (MMPs) [75]. Recently, fucoidan has been shown to inhibit transforming growth factor beta 1 (TGF-β1) secretion and affect cancer cell viability through increased expression of C-type lectin-like receptor 2 (CLEC-2) in several gastric carcinoma cells, demonstrating its anti-cancer potential [92]. However, the role of fucoidan in the progression of intestinal diseases should be clarified by further studies using higher animal models.

## 6. Conclusions and Perspectives

As the intestine is closely associated with nutrient uptake, natural compounds have been studied as potential therapeutic agents to enhance intestinal immunity in humans. Although the effect of fucoidans in cell lines and animal models has been investigated, many questions remain unanswered, such as cellular phenotypes changed by fucoidan intake being a direct trigger to increase humoral immune responses. The structural composition of fucoidans varies with different seaweed species, harvest locations, and extraction methods. In addition, the bioactivity of fucoidans differs based on the degree of sulfation, monosaccharide composition, molecular weight, and the route of administration. Therefore, a better understanding of how fucoidan works in the intestinal area as a dietary additive or food supplement will help develop diet-based therapies for bowel diseases, such as IBDs or colorectal cancers (CRCs).

## Figures and Tables

**Figure 1 marinedrugs-19-00436-f001:**
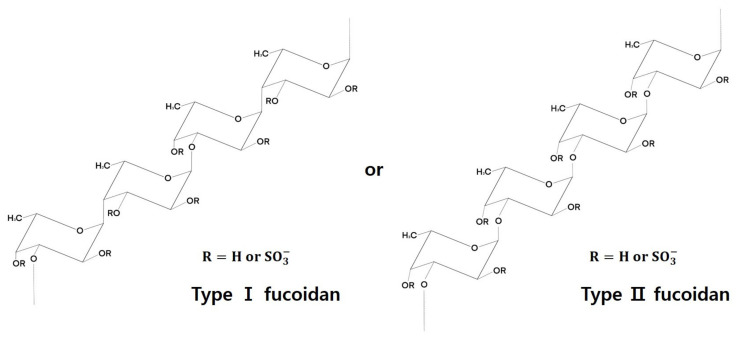
The structure of fucoidan (reproduced from Shang, 2020 [48]).

**Figure 2 marinedrugs-19-00436-f002:**
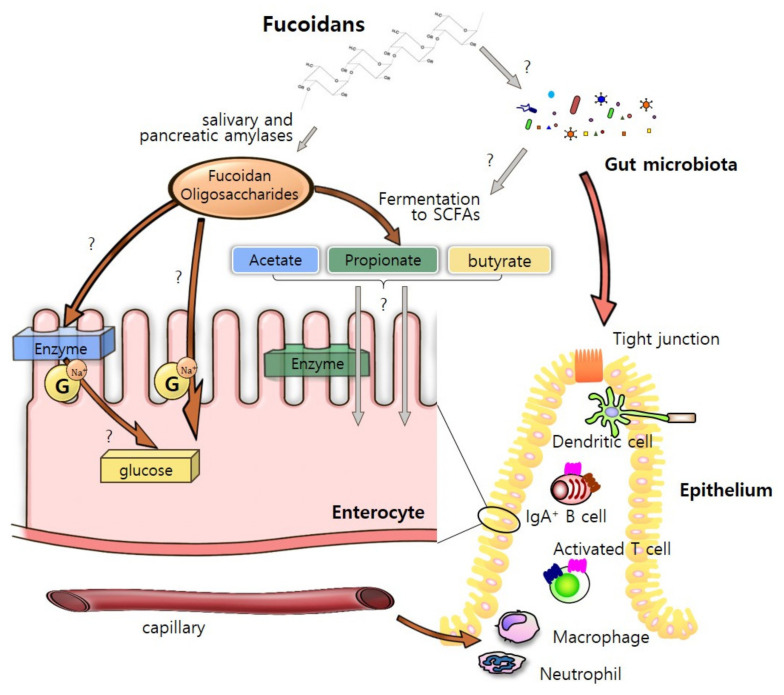
A pivotal role of fucoidans in the intestine. We suggest three possible routes of transporting fucoidans through intestinal epithelium. The fucoidan and its fermenters, short-chain fatty acids (SCFAs), can be digestive by several enzymes resided on intestinal epithelial cells, especially enterocytes. Although the mechanism of nutrient uptake is still unclear, fucoidan can lead to potential effect on host immunological homeostasis and microbiota composition. The ideas of the figure were based on [30,66,67,68,69,70].

**Table 1 marinedrugs-19-00436-t001:** Effects of fucoidan extracts on non-immune cells and immune cells.

Cells	Fucoidan Sources	Fucoidan Concentration Ranges	Positive/Negative Controls	Results	References
Caco-2	*Sargassum cinereum*	0–1000 μg/mL	fluorouracil (a standard drug, 4 μg/mL) as positive control intact cells as a negative control	increase in ROS production and augment mitochondrial membrane permeability decrease in rate of viable cells with increasing fucoidan concentrations enhance ROS induced apoptosis	[25]
Human enterocyte-like HT-29-luc cells	*Undaria pinnatifida*	0–11.1 mg/mL	intact cells as a negative control DSS-induced group with H_2_O_2_ treatment as a negative control	higher superoxide anion (O2^−^) radical scavenging capacities increased survival rate of the cells under H_2_O_2_ toxicity	[26]
Caco-2	*Laminaria japonica*	1.0 mg/mL	intact cells as a negative control	reductions in NO, TNF-α and IL-6 productions. decreased intestinal paracellular permeability of fluorescein isothiocyanate-dextran.	[27]
IECs	*Undaria pinnatifida*	0–100 μg/mL	unstimulated mice as a negative control infected mice by *Cryptosporidium parvum* as a positive control	reduction in binding of *Cryptosporidium parvum* oocysts to the cells	[28]
Intestinal macrophages	*Cladosiphon okamuranus*	0.1–2.0 mg/mL	rats fed on standard chow as a negative control	identified ED1 (macrophage marker) positive cells internalizing fucoidans	[30]
RAW 264.7 cells	*Fucus vesiculosus*	0–200 μg/mL	intact cells as a negative control	reductions in NO, PGE2, TNF-α and IL-1β productions recruitment of macrophages and neutrophils	[31]
Bone marrow-derived DCs	*Fucus vesiculosus*	0–600 mg/kg	unstimulated NOD mice as a negative control	lower expression levels of MHCII and CD86	[32]
Spleen-derived B cells	*Cladosiphon novae-caledonias Kylin* and *Fucus vesiculosus*	0–100 μg/mL	intact cells as a negative control	decrease in NF-κB p52 in B cells, stimulated murine B cell proliferation for 48 h incubation after co-stimulation with interleukin IL-4 and anti-CD40 antibodies	[33]
Spleen-derived cytotoxic T cells	*Okinawa mozuku*	0–3 × 10^5^ g/moL	A mice group fed control diet	increase in CD8^+^ T cells decrease in the ratio of CD4^+^/CD8^+^ T cells	[34]
Blood-derived T cells	*Laminaria Japonica*	0–500 µg/mL	intact cells as a negative control	larger Th1 and Treg populations increase in IL-10 production	[35]
Spleen-derived DCs and T cells	*Fucus vesiculosus*	0–10 mg/kg	intact cells as a negative control	increases in CD40, CD80, CD86, IL-6, IL-12 and TNF-α in spleen DCs promoted the generation of IFN-γ-producing Th1 and Tc1 cells in an IL-12-dependent manner up-regulation of MHC class I and II on spleen cDCs and strongly prompted the proliferation of CD4 and CD8 T cells	[36]
Human mononuclear U937 cells	*Fucus vesiculosus* L.	0–0.25 μg/mL	intact cells (no stimulation with LPS) as a negative control control cells stimulated with LPS (1 μg/mL) as a negative control SB203580 (p38 MAPK Inhibitor, 1.88 μg/mL) as positive control	higher inhibition of MAPK p38 than those of SB203580 a greater inhibition of the COX-2 enzyme with a higher selectivity index than the synthetic non-steroidal anti-inflammatory drug indomethacin higher amount of fucoidan increased the thromboplastin and thrombin time	[37]

**Table 2 marinedrugs-19-00436-t002:** Summary of the fucose-contained brown seaweeds.

Brown Seaweed spp.	Order	Fucose Residue	Backbone	Molecular Weight (kDa)	Weight Ratio of Basic Sugars	References
*Fucus vesiculosus*	Fucales	2-*O*-sulfated fucose 2,3-di-*O*-sulfated fucose	α(1→3) and α(1→4) linked fucose	20–200	Fuc:Gal:Man = 1.00:<0.02:<0.02	[49,50]
*Ascophyllum nodosum*	Fucales	2-*O*-sulfated fucose 2,3-di-*O*-sulfated fucose	α(1→3) and α(1→4) linked fucose	417	Fuc:Xyl:Gal = 31.5:2:3	[50]
*Fucus evanescens*	Fucales	2-*O*-sulfated fucose 2,4-di-*O*-sulfated fucose	α(1→3) and α(1→4) linked fucose	620	Fuc:Glc:Gal:Man:Xyl:Rha = 81:3:4:2:8:2	[51,52]
*Fucus serratus*	Fucales	2-*O*-sulfated fucose	α(1→3) and α(1→4) linked fucose	1705	Fuc:Xyl:Gal:Man:Glc = 54.8:4.0:2.6:1.4:0.6	[42]
*Sargassum linifolium*	Fucales	residues of D-galactose, D-xylose, and L-fucose with sulfate attached to some galactose and fucose residues	a backbone chain of D-glucuronic acid and D-mannose residues	-	Fuc:Xyl:Gal = 1.04:1.00:1.12	[53,54]
*Sargassum stenophyllum*	Fucales	2,4-di-*O*-sulfated fucose	α(1→3) and α(1→4) linked fucose	-	Fuc:Xyl:Man:Gal = 67.8:16.1:1.2:13.6	[55]
*Sargassum fusiforme*	Fucales	complicated glycosyl linkages, including terminal, 1,3-, 1,4-, 1,2,3-, and 1,3,4-linked Fucp, largely due to sulfate substitution at different positions	α(1→2) linked α-d-Man*p* and α(1→4) linked β-d-Glc*p*A	224	Fuc:Xyl:Man:Gal:Rha:Glc:Fru = 28.8:3.9:6.0:12.3: 2.3:1.0:12.3	[56,57]
*Chorda filum*	Laminariales	2,4-di-*O*-sulfated fucose 4-*O*-sulfated fucose	α(1→2) and α(1→3) linked fucose	10–1000	Fuc:Xyl:Man:Glc:Gal = 1.00:0.14:0.15:0.40:0.10	[44]
*Laminaria saccharina*	Laminariales	2,4-di-*O*-sulfated fucose 4-*O*-sulfated fucose	α(1→2) and α(1→3) linked fucose	-	-	[58]
*Lessonia vadosa*	Laminariales	2,4-di-*O*-sulfated fucose 4-*O*-sulfated fucose	α(1→3) linked fucose	320	Fuc:Sulfate = 1.0:1.12	[59]
*Ludwigothurea grisea*	Echinodermata	2-*O*-sulfated fucose 2,4-di-*O*-sulfated fucose	α(1→3) linked fucose	40	Fuc:Gal:Glu:Sulfate = 13.9:1.0:0.5:13.9	[60,61]
*Cladosiphon okamuranus*	Chordariales	4-*O*-sulfated fucose a portion of the fucose residues is O-acetylated at C-5	α(1→2) and α(1→3) linked fucose	75	Fuc:Glu:Sulfate = 6.1:1.0:2.9	[41]
*Adenocystis utricularis*	Ectocarpales	4-*O*-sulfated fucose	mainly composed of 3-linked α-l-fucopyranosyl backbone	6.5–19	Fuc:Man:Glc:Gal = 74:2:1:22	[62]
*Analipus japonicas*	Ectocarpales	2-*O*-sulfated fucose 2,4-di-*O*-sulfated fucose	mainly α(1→3) linked fucose	-	Fuc:Sulfate = 3:2	[63]

**Table 3 marinedrugs-19-00436-t003:** Effects of dietary fucoidan as an immune regulator.

Species	Fucoidan Sources	Dose	Tissues	Results	References
Human	*Fucus vesiculosus* and *Undaria pinnatifida*	1 g/d	feces	increase in fecal lysozyme	[71]
C57BL/6 mice	*Fucus vesiculosus*	5 mg/mL	colon, spleen, and feces	reduced diarrhea and fecal blood loss lower in colon and spleen weight decreases in IL-1α, IL-1β, IL-10, MIP-1α, MIP-1β, G-CSF and GM-CSF in the colon tissue	[72]
Newly weaned pig	*Laminaria* spp.	240 ppm	colon	increased intestinal villous height and the ratio of villus height to crypt depth	[73]
Nile tilapia	Fucus vesiculosus	0.1% 0.2%, 0.4%, or 0.8% in basal diet	intestine	improved WG and SGR increases in organosomatic index in the intestine higher IEL and IEC counts	[74]
Newly weaned pig	*Laminaria* spp.	2.8 g/kg	feces	increased the coefficient of total tract apparent digestibility decreased counts of *Escherichia coli*	[75]
C57BL/6J mice	*Ascophyllum nodosum*	400 mg/kg	colon	decreases in TNF-α, IL-1β, IL-6, and IL-10	[76]
C57BL/6 mice	*Fucus vesiculosus*	400 mg/kg	Colon and feces	reduction in diarrhea and fecal blood decreased the infiltration of inflammatory cells and the expression levels of TNF-α and IL-1β in the colon tissue	[72]

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
