# Peer review of "Fucoidans and Bowel Health"

_marinedrugs, 2021, doi:10.3390/md19080436_

Round 1

Reviewer 1 Report

Fucoidan is one of the key algal polysaccharides with potential applications in health. The review covers an emerging area of its health benefit - the gut health. The review structure was well framed, discussions were focused, and the scope appeared to be well covered. I would recommend to accept after the authors address the following minor edits/questions/comments.

Page 1, line 43: effects, should it be efforts?

Page 2, lines 93-97: discussion on laminarin can be removed, as this is focused on fucoidan  

Page 4, line 156: fucose-containing

Page 5, Figure 1: in the top of the figure, fucoidan was shown to be depolymerized by salivary and pancreatic amylases to fucoidan oligosaccharides. This may need to carefully considered, as most fucoidans are not hydrolyzed by amylases.

Author Response

Fucoidan is one of the key algal polysaccharides with potential applications in health. The review covers an emerging area of its health benefit - the gut health. The review structure was well framed, discussions were focused, and the scope appeared to be well covered. I would recommend to accept after the authors address the following minor edits/questions/comments.

Page 1, line 43 (59): effects, should it be efforts?

Page 2, lines 93-97 (109-113): discussion on laminarin can be removed, as this is focused on fucoidan  

Page 4, line 156 (172): fucose-containing

Page 5, Figure 1: in the top of the figure, fucoidan was shown to be depolymerized by salivary and pancreatic amylases to fucoidan oligosaccharides. This may need to carefully considered, as most fucoidans are not hydrolyzed by amylases.

-> Thank you so much for your valuable comments. We have revised them based on your comments.

Reviewer 2 Report

This review deals with currently very interesting and increasingly important class of marine polysaccharides-fucoidans. The paper is well organised, easy to follow, and the topic is intriguing and well chosen. I recommend this review to be published in Marine Drugs.

Author Response

This review deals with currently very interesting and increasingly important class of marine polysaccharides-fucoidans. The paper is well organised, easy to follow, and the topic is intriguing and well chosen. I recommend this review to be published in Marine Drugs.

-> Thank you so much for your kind review

Reviewer 3 Report

While reading the manuscript, I had some questions and recommendations.

  1. Please provide a link to the literature source of figure 1. Please discuss this figure in terms of the composition of fucoidan, its molecular weight, residues of polyphenols, proteins and other substances.
  2. Supplement Table 1 with molecular weight and weight ratio of basic sugars.
  3. To visualize the data of section 2, please, the data of in vitro tests as table with information on the algae source, cell lines, concentration range, positive/negative controls, effects. Compare the data on fucoidan with the literature, for example (https://doi.org/10.3390/md18050275).
  4. A key aspect of the diet is the absorption of its compounds. Consider absorption of fucoidan, especially in the intestines. Discuss the bioavailability of fucoidan (https://doi.org/10.3390/md16040132, https://doi.org/10.3390/md18110557 etc.) in terms of effects on immunity.
  5. To visualize the data of section 4.2, please, the data of in vivo tests as table with information on the algae source, doses, the route of administration, animals, positive/negative controls, effects.

Author Response

Comments and Suggestions for Authors

While reading the manuscript, I had some questions and recommendations.

  1. Please provide a link to the literature source of figure 1. Please discuss this figure in terms of the composition of fucoidan, its molecular weight, residues of polyphenols, proteins and other substances.

-> Thank you so much for your valuable suggestions. We have revised the section 4.1 based on your comments (line 187-196 and Figure 1 legend).

  1. Supplement Table 1 with molecular weight and weight ratio of basic sugars.

-> We have added the information in Table 2.

  1. To visualize the data of section 2, please, the data of in vitro tests as table with information on the algae source, cell lines, concentration range, positive/negative controls, effects. Compare the data on fucoidan with the literature, for example (https://doi.org/10.3390/md18050275).

-> We have added new table (please see the Table 1)

  1. A key aspect of the diet is the absorption of its compounds. Consider absorption of fucoidan, especially in the intestines. Discuss the bioavailability of fucoidan (https://doi.org/10.3390/md16040132, https://doi.org/10.3390/md18110557 etc.) in terms of effects on immunity.

-> we have discussed about bioavailability of fucoidan (Line 115-119, Line 218-225)

  1. To visualize the data of section 4.2, please, the data of in vivo tests as table with information on the algae source, doses, the route of administration, animals, positive/negative controls, effects.

-> We have added new table (please see the Table 3)

 Reviewer 4 Report

The article titled: "Fucoidans and bowel health " written by: Jin-Young Yang , Sun Young Lim could be of interest for Marine Drugs readers, but in my opinion, in its present form, it cannot be published in the paper. As for review, the article (6 pages) is too short and to be published it should contain much more detailed information about the described topic.

Some other comments are given below:'

  1. line 15 - it should be "their"
  2. lines 26, 37, 49 - what do the Authors mean by: "their"
  3. lines 79-80 - "Notable, a few suggest" - please give specific names of researches authors.
  4. lines 159-160 - it is not a sentence - it should be rearranged.
  5. line 164 - it should be: "help in breakdown"
  6. line 168 - a source of Figure 1 should be presented.

Author Response

Comments and Suggestions for Authors

The article titled: "Fucoidans and bowel health " written by: Jin-Young Yang , Sun Young Lim could be of interest for Marine Drugs readers, but in my opinion, in its present form, it cannot be published in the paper. As for review, the article (6 pages) is too short and to be published it should contain much more detailed information about the described topic.

Some other comments are given below:'

  1. line 15 (31) - it should be "their"
  2. lines 26, 37, 49 (42,53,65)- what do the Authors mean by: "their"
  3. lines 79-80 (95-96) - "Notable, a few suggest" - please give specific names of researches authors.
  4. lines 159-160 (175-176) - it is not a sentence - it should be rearranged.
  5. line 164 (180)- it should be: "help in breakdown"
  6. line 168 (184) - a source of Figure 1 should be presented.

-> Thank you so much for your valuable suggestions. We have revised them based on your comments and put more detail information by adding two more tables and citing more studies throughout the MS.

Round 2

Reviewer 3 Report

The authors provide answers to my questions, however I do not see the corresponding corrections in the text (no table 3, no comparison with https://doi.org/10.3390/md18050275, the table1 is not supplemented with information concentration range, positive/negative controls). Please provide a revised version of the manuscript. 

Discussion of diseases of the bowel health without pharmacokinetic data and absorption is incorrect. Please discuss the pharmacokinetic data (for exp. https://doi.org/10.1016/j.ijbiomac.2019.10.018, https://doi.org/10.3390/md16040132, https://doi.org/10.3390/md18110557 ) 

Author Response

  • In these revised manuscript, we provide more detail information based on your comments. (Please see the red letters through MS):
  1. Table 3 was cited in line 206.
  2. We have added more information in Table 1 and line 152-156.
  3. We have discussed about pharmacokinetic and tissue distribution of fucoidan in line 223-233.

Reviewer 4 Report

I would like to thank the Authors for all corrections and modifications done in the article according to Reviewer's comments. The article is interesting, well organized and deals with important topic, but I am still not sure if the article is long enough to be publishes as review article. Review articles are considered to be published with 20-25 pages. Additionally, it will be essentail to present chemical structures of chosen fucoidans in chapter 3.

Author Response

  • We appreciate this reviewer’s constructive comments. We have added a figure of chemical structures of fucoidan (Figure 1). We will consider the paper as a mini-review article.

Round 3

Reviewer 3 Report

The authors have made the necessary corrections and I have no more questions.